# SMARCD3 Promotes Epithelial–Mesenchymal Transition in Gastric Cancer by Integrating PI3K-AKT and WNT/β-Catenin Pathways

**DOI:** 10.3390/cancers17213526

**Published:** 2025-10-31

**Authors:** Ji-Ho Park, Sun Yi Park, Eun-Jung Jung, Young-Tae Ju, Chi-Young Jeong, Ju-Yeon Kim, Taejin Park, Miyeong Park, Young-Joon Lee, Sang-Ho Jeong

**Affiliations:** 1 Department of Surgery, Gyeongsang National University Hospital, Jinju 52727, Republic of Korea; 2College of Medicine, Gyeongsang National University, Jinju 52727, Republic of Koreadrjej@gnu.ac.kr (E.-J.J.);; 3Department of Surgery, Gyeongsang National University Changwon Hospital, Changwon 51472, Republic of Korea; 4Department of Anesthesiology, Gyeongsang National University Changwon Hospital, Changwon 51472, Republic of Korea

**Keywords:** gastric neoplasm, biomarker, epithelial–mesenchymal transition, prognosis

## Abstract

This study demonstrates that SMARCD3 overexpression in gastric cancer (GC) serves as a negative prognostic biomarker, correlating with significantly poorer overall survival. Mechanistically, high SMARCD3 expression drives tumor aggressiveness by promoting the Epithelial–Mesenchymal Transition (EMT) phenotype. This promotion involves the upregulation of cellular migration and invasion, which is linked to the activation and crosstalk of key oncogenic pathways, specifically the PI3K-AKT (p-AKT-S473, PI3Kp85) and WNT/β-catenin axes. These results identify SMARCD3 as a critical potential therapeutic target for EMT-driven GC progression.

## 1. Introduction

SWI/SNF chromatin remodeling complexes play a pivotal role in regulating gene expression by modulating chromatin accessibility and thereby influencing various cellular processes, including differentiation, proliferation, and tumorigenesis [1,2]. SMARCD3 (SWI/SNF-related, matrix-associated, actin-dependent regulator of chromatin subfamily D member 3) is a key subunit of the SWI/SNF complex that functions as an epigenetic modulator by altering chromatin structure and facilitating transcription factor access [3,4]. Emerging evidence has implicated SMARCD3 as a critical oncogenic driver in various cancers, including gastric, colorectal, and pancreatic cancers [5,6]. Its overexpression is correlated with enhanced epithelial–mesenchymal transition (EMT) and increased cell proliferation, migration, and invasion [7]. Functionally, SMARCD3 acts as an epigenetic modulator and scaffold that integrates extracellular cues with chromatin remodeling, thereby sustaining oncogenic transcriptional programs and cooperating with major signaling pathways, including the PI3K-AKT, WNT/β-catenin, MAPK, and TGF-β pathways, to promote the nuclear translocation and transcriptional activation of EMT-related factors, including Snail, β-catenin, and c-Myc [7,8,9,10,11,12,13].

Recent studies regarding pancreatic cancer have revealed that SMARCD3 amplification in cancer stem cells and its ability to control metabolic and fatty acid pathways are linked to therapeutic resistance [14]. Furthermore, in the colorectal cancer microenvironment, SMARCD3 expression is enriched in cancer-associated fibroblasts (CAFs) and promotes metastasis through positive feedback loops involving WNT5A, TGF-β, and MAPK signaling [15]. Compelling evidence from our group indicates that high SMARCD3 expression serves as a robust negative prognostic factor for gastric cancer patients, correlating with markedly reduced survival rates. Crucially, SMARCD3-mediated epigenetic reprogramming facilitates a pro-metastatic cellular phenotype. In line with this, in vitro functional studies established that SMARCD3 upregulation drives heightened cellular migration and invasion, whereas its knockdown dramatically attenuates these aggressive cellular behaviors, including proliferation. Given its central role in coordinating key oncogenic pathways and Epithelial–Mesenchymal Transition (EMT), a deeper understanding of SMARCD3’s function in gastric cancer signaling is therefore critical for identifying novel therapeutic targets [7,11,16]. Despite its known oncogenic role, the precise molecular mechanism by which SMARCD3 epigenetically integrates the PI3K-AKT and WNT/beta-catenin pathways to drive EMT in gastric cancer remains undefined.

The aim of this study is to elucidate the molecular mechanism through which SMARCD3 integrates with the PI3K-AKT and WNT/β-catenin signaling pathways to promote EMT and gastric cancer progression. We hypothesize that SMARCD3 is a vital epigenetic regulator, facilitating the nuclear localization and activity of EMT transcription factors and thereby increasing metastatic potential. Elucidating this signaling axis will advance precision targeting strategies for aggressive gastric cancers characterized by EMT-driven invasiveness.

## 2. Materials and Methods

### 2.1. Cell Lines and Plasmid Transfection

The human gastric carcinoma cell lines MKN45 and MKN74 were provided by the Korean Cell Line Bank (Seoul, Republic of Korea). All the cell lines were verified by short tandem repeat (STR) profiling and routinely tested to exclude mycoplasma contamination. Cells were grown in RPMI 1640 medium (Gibco, Thermo Fisher Scientific, Inc., Waltham, MA, USA) supplemented with 10% heat-inactivated fetal bovine serum (Gibco, Thermo Fisher Scientific, Inc., Waltham, MA, USA) and 100 U/mL penicillin (Thermo Fisher Scientific, Inc., Waltham, MA, USA). The cultures were maintained at 37 °C in a humidified incubator containing 5% CO_2_. For plasmid overexpression, cells were transfected with either a pCMV6-entry empty vector (cat #: PS100001; OriGene Technologies, Rockville, MD, USA) or a pCMV6-SMARCD3 construct tagged with FLAG-DDK (cat #: RC222004; OriGene Technologies, Rockville, MD, USA) using Lipofectamine 2000 (Invitrogen, Waltham, MA, USA) according to the manufacturer’s protocol. Stable SMARCD3-overexpressing lines were generated in MKN45 and MKN74 cells. Two days following transfection, the cells were placed under selection concentrations of G418 (Geneticin, Cat #: 10131-035; Gibco, Life Technologies Limited, Renfrew, UK) (MKN45 cells: 700 μg/mL, MKN74 cells: 450 μg/mL). Surviving colonies were isolated and expanded, and SMARCD3 expression was validated by Western blot analysis prior to subsequent functional experiments.

### 2.2. LY294002 Treatment

To pharmacologically inhibit the PI3K signaling pathway, cultured cells were treated with LY294002 (Cat #: 440202; EMD Millipore Corp., Carlsbad, CA, USA), a selective inhibitor of PI3K. LY294002 stock solutions were prepared in dimethyl sulfoxide (DMSO) and diluted to the working concentration immediately before use. For downstream analyses, including Western blot (WB) and immunofluorescence (IF) analyses, cells were exposed to LY294002 at a final concentration of 20 µM precisely 24 h prior to harvest. The control groups received an equivalent volume of the DMSO vehicle. Following LY294002 treatment, cells were collected for WB, IF, and nuclear–cytoplasmic fractionation assays to evaluate pathway modulation and epithelial–mesenchymal transition (EMT) marker expression. All experiments were repeated at least three times.

### 2.3. Western Blot Analysis

Cells were harvested and lysed in RIPA buffer (Cat #: 89901; Thermo Fisher Scientific, Inc.) supplemented with protease inhibitor cocktail (Cat #: P8340; Sigma, St. Louis, MO, USA) and phosphatase inhibitor cocktail (Cat #: 78420; Thermo Fisher Scientific, Inc.). Protein lysates were quantified using the Bradford protein assay (Bio-Rad, Hercules, CA, USA), and equal amounts of total protein (20 μg) were resolved on 10% SDS–PAGE gels and transferred onto PVDF membranes using an iBlot 2 Dry Blotting System (Invitrogen). The membranes were blocked with 5% skim milk in TBS-T buffer (20 mM Tris-HCl, 150 mM NaCl, 0.1% Tween-20; pH 7.6) for 1 h at room temperature and incubated overnight at 4 °C with primary antibodies against E-cadherin (Cat #: sc-21791; Santa Cruz Biotechnology Inc., Dallas, TX, USA), SMARCD3 (Cat #: 62265; Cell Signaling Technology, Danvers, MA, USA), Snail (Cat #: ab216347; Abcam, Waltham, MA, USA), Slug (Cat #: ab51772; Abcam, Waltham, MA, USA), and GAPDH (Cat #: sc-47724; Santa Cruz Biotechnology, Dallas, TX, USA; used as loading controls). After three washes in TBS-T, the membranes were incubated with the corresponding HRP-conjugated secondary antibodies (Thermo Fisher Scientific) for 1 h at room temperature. Detection was performed using an ECL chemiluminescent substrate (Bio-Rad, Hercules, CA, USA), and signals were visualized with a ChemiDoc MP Imaging System (Bio-Rad, Hercules, CA, USA). All experiments were repeated at least three times.

### 2.4. Immunocytochemistry

Cells were fixed in 4% paraformaldehyde (PFA) in phosphate-buffered saline (PBS) for 10 min at room temperature. The cells were permeabilized with 0.5% Triton X-100 in PBS for 10 min, followed by blocking in 1% bovine serum albumin (BSA) in PBS for 1 h to minimize nonspecific binding. The cells were incubated overnight at 4 °C with the following primary antibodies diluted in PBS containing 5% BSA: rabbit anti-SMARCD3 (Catalog #: 720131; Thermo Fisher Scientific, Inc., 1:200 in PBS; 5% BSA), mouse anti-E-cadherin (Cat #: sc-21791; Santa Cruz Biotechnology Inc., 1:200 in PBS; 5% BSA), rabbit anti-β-catenin (Cat #: 8480; Cell Signaling Technology, 1:200 in PBS; 5% BSA), and mouse anti-Snail (Catalog #: 14-9859-82; Thermo Fisher Scientific, Inc., 1:200 in PBS; 5% BSA). After three washes with PBS, the cells were incubated with the appropriate fluorescence-conjugated secondary antibodies for 1 h at room temperature. After three additional washes, the nuclei were counterstained with DAPI using ProLong™ Gold Antifade MountantI (Cat #: p36930; Invitrogen, Eugene, OR, USA). Fluorescence images were captured with a Nikon Eclipse Ti confocal microscope (Nikon Instruments Inc., Tokyo, Japan) using identical imaging settings across all the samples to ensure comparability.

#### Microscopic Assessment of Cell Morphology by Microscopy

Cell morphology was monitored in stably SMARCD3-overexpressing MKN74 cells. Differences in cell size and spreading were measured in high-resolution fields using a Nikon Eclipse Ti-S inverted microscope (Nikon Instruments, Inc.). At least three independent images from randomly chosen fields were analyzed for each condition. The mean cellular area was quantified over a range of 50–100 μm using ImageJ software 1.52d, and datasets from the control and SMARCD3-expressing groups were compared.

### 2.5. Statistical Analysis

All the statistical analyses were performed using SPSS Statistics version 27 (IBM Corp., Armonk, NY, USA) and GraphPad Prism version 8.0 (GraphPad Software Inc., San Diego, CA, USA). The chi-square (χ^2^) test was applied for categorical variables. Continuous variables were compared between two independent groups using Student’s *t* test. For in vitro assays, quantitative data were analyzed using GraphPad Prism. Statistical significance was defined as a two-sided *p* value < 0.05.

### 2.6. Ethics Approval

All the experimental procedures conformed to the ethical standards of the Declaration of Helsinki (2013 version). The study protocol was reviewed and approved by the Institutional Review Board of GNUH (approval number: GNUHIRB 2009-54).

## 3. Results

### 3.1. NGS Analysis and Pathway Network

Next-generation sequencing (NGS) was performed after the SMARCD3 gene was overexpressed in the MKN45 and MKN74 gastric cancer cell lines. The pCMV6-entry vector served as the control treatment for each cell line, while the SMARCD3-overexpressing genes #1 and #2 served as the experimental groups (Figure 1A). Multidimensional scaling (MDS) of next-generation sequencing (NGS) data revealed similarities or differences in samples based on gene expression or genetic profiles in MKN45 and MKN74 cells, comparing the control group with the SMARCD3 overexpression group (Figure 1B). In the gene analysis, this represents the number of significant expression values that increased or decreased by more than twofold. In MKN45 cells, 426 genes showed upregulated expression, and 396 genes showed downregulated expression in the SMARCD3 overexpression group. In MKN74 cells, 692 genes showed upregulated expression, and 740 genes showed downregulated expression. Across both cell lines, 68 genes showed a significant increase in expression of more than twofold, while 166 genes showed a significant decrease (Figure 1C). We observed that increased expression of SMARCD3 correlated strongly with the upregulation of key gene expression involved in both the PI3K-AKT pathway and the WNT pathway (Figure 1G) as well as an enhanced epithelial–mesenchymal transition (EMT) gene signature. These data suggest the functional integration of these pathways in driving EMT processes (Figure 1D–G).

### 3.2. Verification of Signal Transduction Pathways Related to Gastric Cancer Cell Line Metastasis

Western blot analysis was performed to compare the expression of representative metastatic markers, including E-cadherin, Snail, and Slug, between the control group (pCMV6-entry) and the stable SMARCD3-overexpressing groups (pCMV6-SMARCD3 #1 and #2) in MKN45 and MKN74 cells. In MKN45 cells, Snail protein expression was markedly increased in the SMARCD3-overexpressing groups, with a significant increase observed in the levels in clone #1 (**** *p* < 0.0001) and clone #2 (* *p* = 0.011) compared with those in the control (Figure 2A and Appendix A). Similarly, in MKN74 cells, SMARCD3 overexpression resulted in an increase in Snail expression in SMARCD3-overexpressing group #1 relative to the control (Figure 2C). LY294002 is a potent and reversible inhibitor of phosphoinositide 3-kinase (PI3K), a key enzyme in the PI3K/AKT signaling pathway. This pathway regulates processes such as cell growth, survival, proliferation, and metabolism. By treating genes associated with EMT with LY294002, we aimed to elucidate its effects on proteins such as the EMT transcription factors Snail, Slug, and E-cadherin. In both MKN45 and MKN74 cells, the increase in Snail protein expression observed in the SMARCD3-overexpressing groups was attenuated upon treatment with LY294002. In particular, a significant reduction in Snail protein levels was detected in MKN45 SMARCD3-overexpressing group #1 (** *p* = 0.009) following LY294002 treatment (Figure 2B and Appendix A). In MKN74 cells, SMARCD3 overexpression tended to increase Snail expression (Figure 2C and Appendix A), whereas this upregulation decreased after LY294002 exposure (Figure 2D and Appendix A).

Given that SMARCD3 has been implicated in the regulation of epithelial–mesenchymal transition (EMT), we next sought to investigate the signaling pathways potentially involved in this process. The PI3K/AKT pathway has been widely reported to promote EMT through downstream effectors such as GSK-3β (ser9) and β-catenin, which regulate transcriptional repressors, including Snail and Slug. In addition, signaling cascades within the MAPK family, including the ERK, p38, and JNK pathways, play crucial roles in modulating cellular proliferation, survival, and invasion during cancer progression. To clarify whether SMARCD3 exerts its EMT-promoting effects through these signaling networks, we examined the expression and phosphorylation status of key molecules—PI3K p85, AKT (Ser473), GSK-3β (Ser9), β-catenin, ERK, p38, and JNK—by Western blotting. In MKN45 cells treated with LY294002, a significant decrease in the expression of these proteins was observed in the control group, with * *p* = 0.0314, while SMARCD3 overexpression group #1 also exhibited a significant reduction in protein expression, with * *p* = 0.02. Furthermore, phosphorylation of p-GSK3β at Ser9 was significantly lower in SMARCD3-overexpressing group #1 than in the control group, with * *p* = 0.015; in group #2, phosphorylation levels also significantly decreased (**** *p* < 0.0001). Additionally, the phosphorylation levels of β-catenin and PERK were lower in the LY294002-treated groups than in the untreated control group (Figure 2E and Appendix A). In MKN74 cells treated with LY294002, the phosphorylation levels of AKT in the control group significantly decreased, with * *p* = 0.0135. The SMARCD3 overexpression groups demonstrated further significant decreases in phosphorylation levels, with group #1 showing ** *p* = 0.0019 and group #2 showing * *p* = 0.0144. With respect to the phosphorylation of p-GSK3β at Ser9, the control group exhibited a significant decrease, with * *p* = 0.007. Even more pronounced decreases were detected in the SMARCD3-overexpressing groups, with group #1 having **** *p* < 0.0001 and group #2 showing * *p* = 0.024. Additionally, the phosphorylation levels of β-catenin, PERK, p-p38, and p-JNK were lower in the LY294002-treated groups than in the untreated control group (Figure 2F and Appendix A). These results suggest that SMARCD3 overexpression suppresses the nuclear accumulation of key EMT-related transcription factors and signaling proteins under PI3K inhibition.

### 3.3. Regulation of Nuclear Translocation of EMT-Related Proteins

The overexpression of SMARCD3 resulted in increased nuclear accumulation of EMT-associated transcription factors, including Snail and c-Myc, concomitant with increased EMT phenotypes. Importantly, treatment with LY294002 diminished these effects, confirming that SMARCD3-induced EMT relies on PI3K-AKT pathway activity. Similarly, nuclear translocation of β-catenin and activation of EMT transcription factors were attenuated. These findings indicate that the PI3K-AKT axis cooperates with WNT/β-catenin signaling to regulate EMT induction in gastric cancer patients. To precisely analyze the nuclear translocation of signaling proteins and transcription factors during the EMT process, changes in the localization of Snail, β-catenin, c-Myc, and Cyclin D were observed. In MKN45 cells treated with LY294002 and separated into nuclear and cytosolic fractions, Snail levels significantly decreased in SMARCD3-overexpressing group #1, with **** *p* < 0.0001, and in group #2, Snail levels significantly decreased, with * *p* = 0.0064. Additionally, c-Myc expression was significantly decreased in SMARCD3-overexpressing group #2, with * *p* = 0.02 (Figure 3A and Appendix A). In the SMARCD3 overexpression group #2 treated with ly294002 in MKN74 cells, Snail levels were significantly reduced at *p* = 0.0008 (Figure 3B and Appendix A). Collectively, these findings indicate that SMARCD3 overexpression promotes the nuclear accumulation of Snail via the PI3K/Akt signaling axis, thus facilitating EMT and contributing to gastric cancer progression and poor prognosis.

### 3.4. Protein Distribution According to EMT Status

To evaluate the role of SMARCD3 in epithelial–mesenchymal transition (EMT), cell number and size were quantified using optical microscopy in three randomized areas of equal unit area per sample. SMARCD3 overexpression significantly increased cell numbers compared with those in the control group (control vs. SMARCD3 overexpression group #1: ** *p* = 0.0035; group #2: * *p* = 0.015). The count of the control cells was 28.67 ± 6.03 cells (mean ± standard deviation), which increased marginally to 35.33 ± 5.13 cells upon LY294002 treatment (*p* = 0.29). The number of SMARCD3-overexpressing cells decreased (13 ± 2.65 in group #1; 16 ± 1 in group #2), but it increased after LY294002 treatment, though this increase did not reach statistical significance (20.67 ± 1.53, *p* = 0.19 in group #1; 25.33 ± 5.86, *p* = 0.08 in group #2). The cell area was significantly larger in the SMARCD3-overexpressing groups than in the control group (**** *p* < 0.0001). LY294002 treatment significantly reduced the cell area both in the control group (from 1184 ± 412.6 to 1024 ± 338.7, *p* = 0.0048) and in the SMARCD3-overexpression groups (from 2442 ± 799.2 to 1392 ± 383.8, ** *p* < 0.0001 in group #1; from 1928 ± 562.5 to 1274 ± 563, *p* < 0.05 in group #2) (Figure 4A). Furthermore, fluorescence microscopy revealed that compared with the control, SMARCD3 overexpression disrupted morphology related to the EMT marker E-cadherin, resulting in less compact and irregular patterns in groups #1 and #2. LY294002 treatment restored the continuity and localization of E-cadherin, restoring it to an intact morphology (Figure 4B). Next-generation sequencing (NGS) analysis demonstrated the activation of the WNT and PI3K-AKT signaling pathways upon SMARCD3 overexpression. To investigate the associated cellular changes, the colocalization of β-catenin and Snail was examined. Compared with control cells, SMARCD3-overexpressing cells (groups #1 and #2) presented increased nuclear localization of Snail, which was reduced following LY294002 treatment, whereas the changes in β-catenin localization were less pronounced. (Figure 4C). Taken together, these findings support a regulatory role for SMARCD3 in modulating EMT through the PI3K-AKT and WNT/β-catenin signaling pathways in gastric cancer cells.

### 3.5. WNT3A Synergistically Boosts SMARCD3-Mediated EMT

To confirm the activation of the WNT pathway and the association between SMARCD3 and epithelial–mesenchymal transition (EMT), WNT signaling was stimulated using WNT3A. Following overexpression of smarcd3 in MKN45 and MKN74 cells, TCF/LEF reporter analysis was performed 4 h after WNT3A treatment. This confirmed that smarcd3 alone also increased β-catenin activity in smarcd3-overexpressing cells (S10). To validate β-catenin and EMT activation in smarcd3-overexpressing cells, cells were fractionated into nuclear and cytoplasmic fractions after WNT3A treatment (final concentration 50 ng/mL, harvested exactly 4 h prior). Changes in the expression of EMT-related proteins β-catenin, c-Myc, and Cyclin D were investigated. In MKN45 cells, nuclear Snail protein expression was significantly higher in the SMARCD3-overexpressing group #1 compared to the control group (Figure 5A and Appendix A; **** *p* < 0.0001). Similarly, MKN74 cells in SMARCD3 overexpression groups #1 and #2 exhibited significantly increased nuclear Snail levels compared to control cells (Figure 5B and Appendix A; ** *p* = 0.001 and ** *p* = 0.002). These results suggest that SMARCD3 may promote the β-catenin–TCF/LEF complex, but whether it acts directly as a coactivator for transcription factors requires additional data, such as luciferase assays, under conditions of reduced β-catenin (KD/KO).

## 4. Discussion

Our findings establish SMARCD3 as a pivotal epigenetic regulator that drives an aggressive gastric cancer phenotype through the orchestration of epithelial–mesenchymal transition (EMT). We demonstrated that SMARCD3 acts as a novel signaling amplifier through the newly defined SMARCD3–PI3K-AKT–WNT axis, which offers a highly promising therapeutic strategy to suppress EMT-driven invasion and combat metastasis [7]. Mechanistically, SMARCD3 overexpression leads to the upregulation of both PI3K-AKT signaling and WNT signaling, which synergistically drives the nuclear accumulation of EMT-related transcription factors such as Snail and Slug. This oncogenic crosstalk is crucially maintained by the inhibitory phosphorylation of GSK3β at Ser9, a crucial modification that stabilizes Snail and promotes EMT progression. Furthermore, WNT3A-induced WNT pathway activation synergizes with SMARCD3 to enhance EMT progression, underscoring the complex interplay of signaling cascades in gastric tumor biology. The elucidation of this axis reveals a specific druggable vulnerability at GSK3β, as the EMT phenotype and nuclear Snail accumulation were entirely reversed upon treatment with the PI3K inhibitor LY294002. These results strongly suggest that targeting upstream PI3K-AKT signaling is an effective strategy to reactivate GSK3β and suppress SMARCD3-driven malignancy [17,18,19]. Furthermore, future translational efforts should focus on highly selective clinical PI3K inhibitors, such as Alpelisib (BYL719), which target the PI3Kα subunit to robustly and specifically restore GSK3β activity and promote EMT factor degradation [20,21].

The identification of the SMARCD3–PI3K-AKT–WNT axis is particularly relevant given the formidable challenges posed by the complex molecular heterogeneity of gastric cancer (GC) and the urgent need for new mechanism-specific targets [22,23]. GC is highly heterogeneous and is categorized using systems such as The Cancer Genome Atlas (TCGA), which reveals distinct subtypes with varying prognoses and therapeutic vulnerabilities [24]. Despite two decades of intense research, outcomes for patients with advanced disease remain largely unsatisfactory, underscoring a pressing need to discover novel molecular targets that are present in a relatively large patient subset [20,25,26,27,28,29].

Current targeted therapies focus on a small percentage of patients and include HER2-targeted agents such as trastuzumab and immune checkpoint inhibitors for microsatellite instability-high (MSI-H) or deficient mismatch repair (dMMR) tumors [22,30]. Additionally, antiangiogenic agents such as ramucirumab are employed as second-line therapy, and the tight junction protein CLDN18.2 has recently emerged as a highly promising gastric cancer-specific target for antibodies such as zolbetuximab [22]. However, the effectiveness of these therapies is often limited by the intrinsic and acquired resistance mechanisms of tumors, which frequently involve hyperactivated survival pathways.

Our findings directly address this therapeutic gap by identifying a mechanism linked to two of the most critical survival pathways. Our findings directly address a significant therapeutic gap in gastric cancer by identifying a novel upstream regulatory mechanism. While the WNT β-catenin and PI3K/AKT/mTOR pathways are well-established drivers of resistance and metastasis, their deregulation often leads to compensatory mechanisms following single-pathway inhibition [16,23]. Defining SMARCD3 as an upstream epigenetic node that synchronously activates both WNT and PI3K/AKT signaling provides a unique and strong rationale for developing strategies that simultaneously suppress both axes [31,32]. This approach promises to effectively combat invasion and metastasis, particularly relevant for aggressive subtypes like diffuse-type GC [16,25].

Despite defining this novel oncogenic axis, our study is subject to several crucial limitations inherent to foundational mechanistic research. First, all functional results were based entirely on in vitro assays; therefore, rigorous in vivo validation in relevant animal models is necessary to fully confirm therapeutic efficacy. Second, the precise upstream epigenetic mechanism by which SMARCD3, a SWI/SNF subunit, alters chromatin to promote the transcription of PI3K and WNT signaling components remains undefined and requires further investigation using techniques such as ChIP-seq. Finally, the prognostic utility of our cell line model requires rigorous clinical validation across large, independent patient cohorts to establish SMARCD3 as a reliable biomarker for aggressive disease and therapeutic stratification.

## 5. Conclusions

SMARCD3 acts as a critical epigenetic regulator that promotes EMT in gastric cancer patients through the integration of PI3K-AKT and WNT/β-catenin signaling. Targeting this SMARCD3-mediated mechanism offers a promising therapeutic strategy to inhibit metastasis and improve outcomes for patients with gastric cancer.

## Figures and Tables

**Figure 1 cancers-17-03526-f001:**
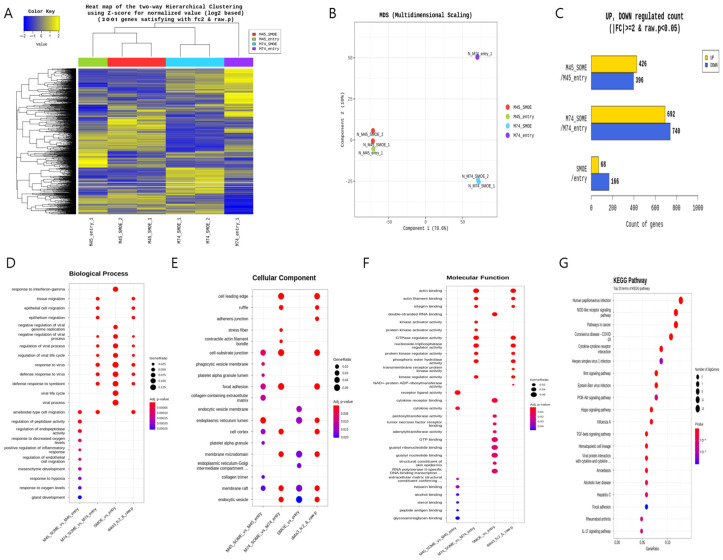
Comparison of the transcriptomes of stable MKN45 and MKN74 cells overexpressing SMARCD3. Heatmap for hierarchical clustering. (**A**) Multidimensional scaling plot. (**B**) Number of genes with significantly upregulated or downregulated expression according to fold change and *p* value. The numbers of genes with upregulated and downregulated expression were compared between the MKN45 and MKN74 cell lines (**C**). Gene Ontology enrichment (BP, CP and MF) of the top module (**D**–**F**) is shown. KEGG pathway enrichment of the top module (**G**).

**Figure 2 cancers-17-03526-f002:**
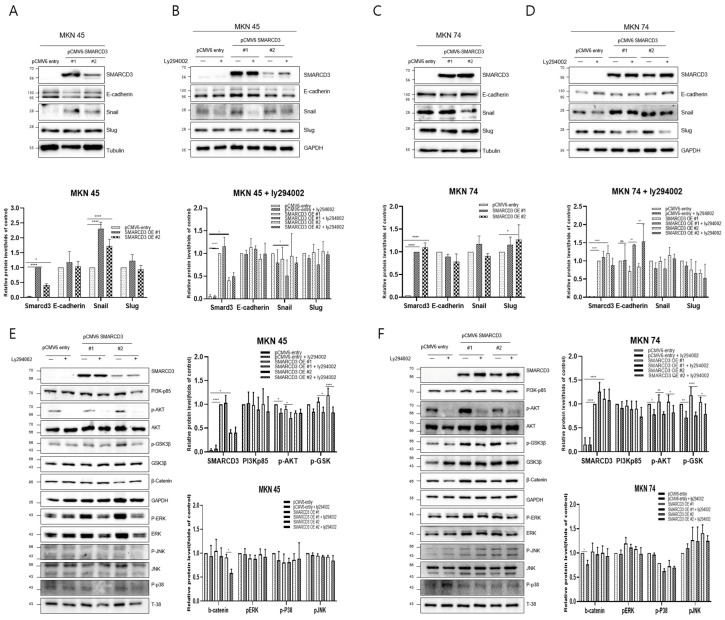
Effects of LY294002 on the expression of EMT markers (**A**–**D**) and signaling proteins (**E**,**F**). Quantitative Western blots revealed decreased phosphorylation of p-GSK3β (Ser9), β-catenin, PERK, p-p38, and p-JNK in LY294002-treated cells compared with those in control cells, with more pronounced effects in the SMARCD3 overexpression groups. The data are presented as the means ± SDs for three independent experiments, and the error bars indicate the SDs. Statistical analysis was performed by two-way ANOVA **** *p* < 0.0001. The original Western blot figures are shown in Appendix A. * *p* < 0.05, ** *p* < 0.01, **** *p* < 0.0001.

**Figure 3 cancers-17-03526-f003:**
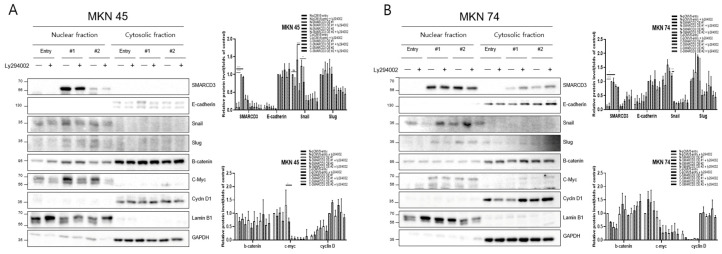
Nuclear and cytosolic fractionation of EMT-related proteins in MKN45 (**A**) and MKN74 (**B**) cells treated with LY294002. Western blot analysis revealed changes in the expression and subcellular localization of SMARCD3, E-cadherin, Snail, Slug, β-catenin, c-Myc, and Cyclin D in nuclear and cytosolic fractions with and without LY294002 treatment. The data are presented as the means ± SDs for three independent experiments, and the error bars indicate the SDs. Statistical analysis was performed by two-way ANOVA. * *p* < 0.05, ** *p* < 0.01, *** *p* < 0.001 **** *p* < 0.0001. The original Western blot figures are shown in Appendix A.

**Figure 4 cancers-17-03526-f004:**
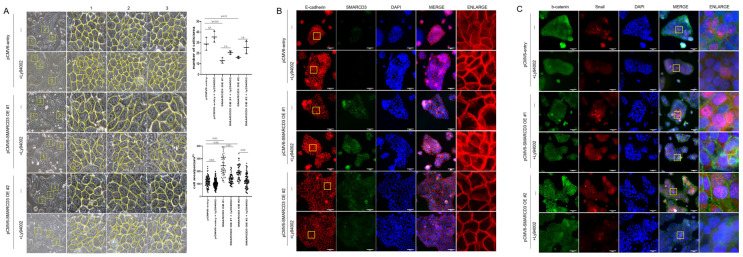
SMARCD3 overexpression in MKN74 cells alters cell morphology and EMT marker distribution, which is reversed by LY294002 treatment. (**A**) Optical microscopy revealed a significant reduction in cell number per unit area and an increase in cell size upon SMARCD3 overexpression; both effects were reversed by LY294002 treatment. Cell number quantification was performed using two-way ANOVA, and cell size was analyzed by Brown–Forsythe and Welch ANOVA tests. Images were acquired at 200× magnification. The yellow box indicates a 200 μm × 200 μm area (mean ± SD). Yellow line outlined the cell morphology. (**B**) Fluorescence microscopy demonstrating the localization of E-cadherin and SMARCD3 in SMARCD3-overexpressing MKN74 cells. (**C**) Immunostaining for β-catenin and Snail; nuclei were counterstained with DAPI. Scale bars represent 50 μm. The magnifications are 200× for (**A**) and 400× for (**B**,**C**).

**Figure 5 cancers-17-03526-f005:**
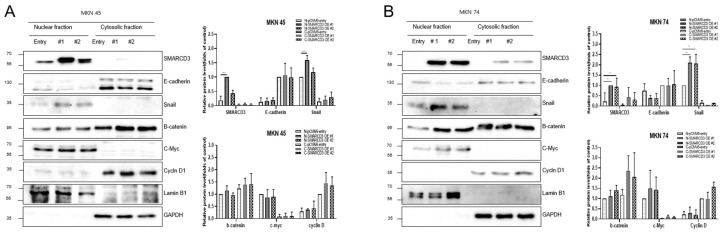
WNT3A treatment activated WNT signaling and increased nuclear Snail accumulation in the groups with SMARCD3 overexpression. Changes in the expression of the EMT-related proteins β-catenin, c-Myc, Cyclin D, and Snail in nuclear and cytosolic fractions were analyzed after WNT3A stimulation in MKN 45 (**A**) and MKN 74 (**B**). The data are presented as the means ± SDs for three independent experiments, and the error bars indicate the SDs. Statistical analysis was performed by two-way ANOVA. ** *p* < 0.01, **** *p* < 0.0001. The original Western blot figures are shown in Appendix A.

## Data Availability

The data presented in this study are available upon request from the corresponding author.

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
