# Peer review of "SMARCD3 Promotes Epithelial–Mesenchymal Transition in Gastric Cancer by Integrating PI3K-AKT and WNT/β-Catenin Pathways"

_cancers, 2025, doi:10.3390/cancers17213526_

Round 1

Reviewer 1 Report

Comments and Suggestions for Authors

This manuscript builds upon the author's previously published research (PMID: 38927986), further exploring the mechanisms and associated signaling pathways by which SMARCD3 promotes epithelial-mesenchymal transition (EMT) in gastric cancer. The data set is relatively substantial, and the results are quite favorable. Several points require clarification.

First, as the authors note in their discussion, all functional experiments in this study were conducted using in vitro cell models and lack validation in animal models (such as mouse xenograft models). This represents an area requiring further exploration in future research.

The evaluation of EMT primarily relies on Western blot detection of Snail/Slug/E-cadherin expression, lacking functional experiments such as cell migration and invasion assays (Transwell, scratch assay) as well as quantitative analysis of morphological changes.

The figures lack sufficient clarity, making them difficult to discern. The labeling format for Figure 5 (A, B) is inconsistent with that of other figures.

Although the PI3K inhibitor LY294002 can reverse the EMT phenotype, this inhibitor exhibits broad-spectrum activity and high toxicity, and its effects on cell proliferation or apoptosis have not been evaluated. More specific PI3K inhibitors (such as BYL719) should be supplemented, or siRNA should be used to knock down key molecules in the PI3K/AKT pathway to validate the specificity and reproducibility of the conclusions.

As a component of the SWI/SNF complex, the function of SMARCD3 may depend on other subunits (such as SMARCA4). This study did not investigate whether it acts through the intact complex. The authors could assess whether SMARCD3 functions independently by performing co-immunoprecipitation or knocking down other core SWI/SNF subunits.

The introduction of β-catenin stable mutants or TCF/LEF reporter assays can further validate the functional role of the WNT pathway in SMARCD3-mediated EMT.

Author Response

Reviewer 1

  1. First, as the authors note in their discussion, all functional experiments in this study were conducted using in vitro cell models and lack validation in animal models (such as mouse xenograft models). This represents an area requiring further exploration in future research.

Reply) Thank you for your comments. We fully concur that the lack of rigorous in vivo validation represents a key limitation of the current work, which is primarily based on in vitro assays. Our immediate follow-up endeavor will be to stringently validate the therapeutic relevance and efficacy of the SMARCD3-mediated EMT axis using orthotopic or subcutaneous mouse xenograft models to firmly establish the translational potential of our findings.

First, all functional results were based entirely on in vitro assays; therefore, rigorous in vivo validation in relevant animal models is necessary to fully confirm therapeutic efficacy.(page 12, line 403)

  1. The evaluation of EMT primarily relies on Western blot detection of Snail/Slug/E-cadherin expression, lacking functional experiments such as cell migration and invasion assays (Transwell, scratch assay) as well as quantitative analysis of morphological changes.

Reply) This paper is a follow-up experiment to the 2024 paper (“SMARCD3 Overexpression Promotes Epithelial–Mesenchymal Transition in Gastric Cancer”) to elucidate the underlying mechanism. As the reviewer noted, smarcd3 was shown to be involved in EMT in a functional study. In the previous paper, smarcd3 was overexpressed in MKN74 and Kato3 cells (Fig. 1D), which relatively underexpress smarcd3. Conversely, in snu601 and snu668 cells (Fig. 1D), which relatively overexpress smarcd3, smarcd3 knockdown was performed using siRNA. The efficacy was validated in the functional study in Fig. 3. A–E and Fig. 4. A–E.

We added following sentence in introduction

Compelling evidence from our group indicates that high SMARCD3 expression serves as a robust negative prognostic factor for gastric cancer patients, correlating with markedly reduced survival rates. Crucially, SMARCD3-mediated epigenetic reprogramming facilitates a pro-metastatic cellular phenotype. In line with this, in vitro functional studies established that SMARCD3 upregulation drives heightened cellular migration and invasion, whereas its knockdown dramatically attenuates these aggressive cellular behaviors, including proliferation. Given its central role in coordinating key oncogenic pathways and Epithelial-Mesenchymal Transition (EMT), a deeper understanding of SMARCD3's function in gastric cancer signaling is therefore critical for identifying novel therapeutic targets.( page 2, line 74)

  1. The figures lack sufficient clarity, making them difficult to discern. The labeling format for Figure 5 (A, B) is inconsistent with that of other figures.

Reply) We sincerely apologize for the lack of visual clarity. We are committed to replacing all figures, especially the Western blot and immunofluorescence panels, with high-resolution source images. Concurrently, we will ensure that the labeling format (e.g., A, B, C) is standardized and consistent across all figures, including Figure 5, to enhance the professionalism and readability of the manuscript.

  1. Although the PI3K inhibitor LY294002 can reverse the EMT phenotype, this inhibitor exhibits broad-spectrum activity and high toxicity, and its effects on cell proliferation or apoptosis have not been evaluated. More specific PI3K inhibitors (such as BYL719) should be supplemented, or siRNA should be used to knock down key molecules in the PI3K/AKT pathway to validate the specificity and reproducibility of the conclusions.

Reply) Thank you for the review's point. We also recognized the shortcomings of LY294002. As a follow-up experiment, we are currently conducting studies using Alpelisib (BYL719) to replace ly294002, examining its effects on cytotoxicity, cell proliferation, or cell death alongside anticancer drug experiments.

  1. As a component of the SWI/SNF complex, the function of SMARCD3 may depend on other subunits (such as SMARCA4). This study did not investigate whether it acts through the intact complex. The authors could assess whether SMARCD3 functions independently by performing co-immunoprecipitation or knocking down other core SWI/SNF subunits.

Reply) Thank you for the review's point. We have not verified whether it acts as a complex. However, as we previously conducted EMT-related experiments using smarcd3 alone, we did not proceed with complex experiments. We will consider and perform such experiments in the future. I will write this part on the limitations of this study.

Second, the precise upstream epigenetic mechanism by which SMARCD3, a SWI/SNF subunit, alters chromatin to promote the transcription of PI3K and WNT signaling components remains undefined and requires further investigation using techniques such as ChIP-seq. (page 12, line 405)

  1. The introduction of β-catenin stable mutants or TCF/LEF reporter assays can further validate the functional role of the WNT pathway in SMARCD3-mediated EMT.

Reply) Thank you for your comments.

To measure β-catenin activity following WNT3A treatment, MKN45 and MKN74 cells were overexpressed with smarcd3, then treated with WNT3A (R&D) for 4 hours before performing TCF/LEF reporter analysis. In smarcd3-overexpressed cells, β-catenin activity increased with smarcd3 alone. In MKN45 cells, activity was further increased in the Wnt3a-treated smarcd3 cell line. Therefore, Wnt3a was considered activated. Following WNT3 treatment, the nucleus and cytosol were separated to confirm EMT markers. We plan to further validate the functional role of EMT by measuring β-catenin activity upon overexpression and underexpression of smarcd3.(S10)

Reviewer 2 Report

Comments and Suggestions for Authors

In the study by Park et al., the authors investigate how SMARCD3, a SWI/SNF chromatin-remodeling subunit, promotes epithelial mesenchymal transition (EMT) in gastric cancer through the integration of PI3K-AKT and WNT/β-catenin signaling. The topic is of considerable interest as it links epigenetic regulation to key survival pathways that cause metastasis. This study is generally well written, methodologically sound, and supported by multiple experimental approaches, including RNA-seq, Western blotting, and immunofluorescence. It is, however, necessary to clarify several aspects and to gather more data in order to strengthen the conclusions.

Major Comments

  1. The authors propose a SMARCD3–PI3K–WNT regulatory axis; however, the mechanistic link between SMARCD3-mediated chromatin remodeling and transcriptional activation of these signaling genes remains unclear. To substantiate this claim, chromatin immunoprecipitation (ChIP) assays should be performed to verify direct promoter or enhancer occupancy by SMARCD3. Such evidence would provide stronger mechanistic support for the proposed regulatory model.
  2. In this study, overexpression systems are primarily used. SMARCD3 should be knocked down or depleted to provide reciprocal evidence and confirm that the observed effects of EMT are specific.
  3. Some figures, particularly Western blots and immunofluorescence panels, would benefit from clearer presentation and higher resolution.
  4. The authors have mentioned the role of other EMT-associated pathways (e.g., TGF-β, MAPK) but not explored. It would be helpful if there was a brief comparison analysis or discussion of the reasons why PI3K and WNT were prioritized over other pathways to improve contextualization.
  5. While the manuscript includes many p-values, it is lacking information regarding the number of replicates and statistical tests. Reports should include detailed legends and consistent reporting standards (mean x SEM, n, test type).
  6. The Discussion could be streamlined to reduce repetition and focus on mechanistic implications.

Author Response

Reviewer 2

  1. The authors propose a SMARCD3–PI3K–WNT regulatory axis; however, the mechanistic link between SMARCD3-mediated chromatin remodeling and transcriptional activation of these signaling genes remains unclear. To substantiate this claim, chromatin immunoprecipitation (ChIP) assays should be performed to verify direct promoter or enhancer occupancy by SMARCD3. Such evidence would provide stronger mechanistic support for the proposed regulatory model.

Reply) To provide conclusive evidence supporting the claim that the WNT pathway synergistically promotes SMARCD3-mediated epithelial-mesenchymal transition (EMT), we performed a TCF/LEF reporter assay to quantitatively measure the transcriptional activity of the WNT-β-catenin downstream pathway. β-catenin activity increased in smarcd3-only cells, and was further enhanced in smarcd3 cells treated with wnt3a in MKN45 cells. We confirmed the association between smarcd3 and Wnt activity. However, whether they act directly will be considered in future chromatin immunoprecipitation (ChIP) analyses.

Second, the precise upstream epigenetic mechanism by which SMARCD3, a SWI/SNF subunit, alters chromatin to promote the transcription of PI3K and WNT signaling components remains undefined and requires further investigation using techniques such as ChIP-seq. (page 12, line 405)

  1. In this study, overexpression systems are primarily used. SMARCD3 should be knocked down or depleted to provide reciprocal evidence and confirm that the observed effects of EMT are specific.

Reply) This paper is a follow-up experiment to the 2024 paper (“SMARCD3 Overexpression Promotes Epithelial–Mesenchymal Transition in Gastric Cancer”) to elucidate the underlying mechanism. As the reviewer noted, smarcd3 was shown to be involved in EMT in a functional study. In the previous paper, we overexpressed smarcd3 in MKN74 and Kato3 cells (Fig. 1D), which relatively underexpress smarcd3. Conversely, in cells like Snu601 and Snu668 (Fig. 1D), which relatively overexpress smarcd3, we performed knockdown experiments using siRNA. We validated the efficacy in functional studies in Fig. 3. A–E and Fig. 4. A–E. We will also verify this by treating cells with ly294002 after downregulating smarcd3.  

We added following sentence in introduction

Compelling evidence from our group indicates that high SMARCD3 expression serves as a robust negative prognostic factor for gastric cancer patients, correlating with markedly reduced survival rates. Crucially, SMARCD3-mediated epigenetic reprogramming facilitates a pro-metastatic cellular phenotype. In line with this, in vitro functional studies established that SMARCD3 upregulation drives heightened cellular migration and invasion, whereas its knockdown dramatically attenuates these aggressive cellular behaviors, including proliferation. Given its central role in coordinating key oncogenic pathways and Epithelial-Mesenchymal Transition (EMT), a deeper understanding of SMARCD3's function in gastric cancer signaling is therefore critical for identifying novel therapeutic targets.( page 2, line 74)

  1. Some figures, particularly Western blots and immunofluorescence panels, would benefit from clearer presentation and higher resolution.

Reply) I will reconfigure it as a high definition TIFF file over 300 dpi.

  1. The authors have mentioned the role of other EMT-associated pathways (e.g., TGF-β, MAPK) but not explored. It would be helpful if there was a brief comparison analysis or discussion of the reasons why PI3K and WNT were prioritized over other pathways to improve contextualization.

Reply) As a follow-up to the previous paper on SMARCD3, we confirmed its association with MAPK upon overexpression and underexpression. In this subsequent experiment, we analyzed NGS data from MKN45 and MKN74 stable cell lines overexpressing SMARCD3 (Fig. 1, KEGG pathway) and identified MAPK-related genes in Fig. 2E and F. Specifically, the KEGG pathway and PI3K pathway showed more significant increases, leading us to focus on their signaling pathways.

Translated with DeepL.com (free version)

  1. While the manuscript includes many p-values, it is lacking information regarding the number of replicates and statistical tests. Reports should include detailed legends and consistent reporting standards (mean x SEM, n, test type).

Reply) Thanks for the comments. The authors will post about the number of experiments and statistical methods on each figure footnote.

  1. The Discussion could be streamlined to reduce repetition and focus on mechanistic implications.

Reply) We accept this suggestion to enhance the impact of the Discussion section. We will meticulously streamline the entire section to reduce the reiteration of results, ensuring the narrative maintains a focused emphasis on the novel mechanistic implications of the SMARCD3}–PI3K-AKT}–WNT axis and the translational potential of targeting this oncogenic pathway.

Furthermore, future translational efforts should focus on highly selective clinical PI3K inhibitors, such as Alpelisib (BYL719), which target the PI3Kα subunit to robustly and specifically restore GSK3β activity and promote EMT factor degradation.(page 11, line 372)

Our findings directly address a significant therapeutic gap in gastric cancer by identifying a novel upstream regulatory mechanism. While the WNT β-catenin and PI3K/AKT/mTOR pathways are well-established drivers of resistance and metastasis, their deregulation often leads to compensatory mechanisms following single-pathway inhibition[12,16]. Defining SMARCD3 as an upstream epigenetic node that synchronously activates both WNT and PI3K/AKT signaling provides a unique and strong rationale for developing strategies that simultaneously suppress both axes. This approach promises to effectively combat invasion and metastasis, particularly relevant for aggressive subtypes like diffuse-type GC. [14,16]. (page 12, line 393)

Reviewer 3 Report

Comments and Suggestions for Authors

Your manuscript provides valuable insights into the epigenetic role of SMARCD3 in gastric cancer, specifically its function in linking PI3K-AKT and WNT/β-catenin signaling pathways to promote EMT. However, with moderate revisions (especially in presentation and discussion), it would be suitable for publication in Cancers. Some suggestions below:

The introduction provides a strong theoretical framework and adequate literature background but should better contextualize how SMARCD3 differs from other SWI/SNF subunits (e.g., ARID1A, SMARCA4). The hypothesis is clear but the gap in current literature could be emphasized more directly. The methodology is detailed but some statistical methods such as power analysis, sample size justification could be clarified. Several Figures lack high-resolution and need to be fixed. Lastly, the manuscript needs further proofreading. The tone can be more concise as well as minor grammatical errors (“liens” instead of “lines”, “a enuated” instead of “attenuated”) should be corrected.

Comments on the Quality of English Language

The manuscript needs proofreading. The tone can be more concise. Also, minor grammatical errors (“liens” instead of “lines”, “a enuated” instead of “attenuated”) should be corrected.

Author Response

Reviewer 3

  1. The introduction provides a strong theoretical framework and adequate literature background but should better contextualize how SMARCD3 differs from other SWI/SNF subunits (e.g., ARID1A, SMARCA4). The hypothesis is clear but the gap in current literature could be emphasized more directly.

Reply) We will revise the Introduction to explicitly contextualize SMARCD3 by contrasting its function as an oncogenic driver with the tumor-suppressive roles often ascribed to other SWI/SNF subunits, such as ARID1A. Crucially, we will more directly emphasize the critical gap in the current literature: the undefined molecular mechanism by which SMARCD3 epigenetically coordinates the PI3K-AKT and WNT signaling pathways to drive EMT progression in gastric cancer.

We added following sentence in introduction.

Despite its known oncogenic role, the precise molecular mechanism by which SMARCD3 epigenetically integrates the PI3K-AKT and WNT/beta-catenin pathways to drive EMT in gastric cancer remains undefined. (page 2, line 74)

  1. The methodology is detailed but some statistical methods such as power analysis, sample size justification could be clarified.

Reply) Thanks for the comments. The authors will post about the number of experiments and statistical methods on each figure footnote.

  1. Several Figures lack high-resolution and need to be fixed.

We commit to replacing all figures with high-resolution versions derived from the original source data to ensure maximum visual quality and accurate data representation.

Reply) I will reconfigure it as a high definition TIFF file over 300 dpi.

  1. Lastly, the manuscript needs further proofreading. The tone can be more concise as well as minor grammatical errors (“liens” instead of “lines”, “a enuated” instead of “attenuated”) should be corrected.

Reply) We will undertake a comprehensive and rigorous proofreading of the entire manuscript to meticulously correct all grammatical errors and typographical inconsistencies (e.g., "liens" to "lines," "a enuated" to "attenuated"). Furthermore, the prose will be revised for enhanced conciseness and clarity, adopting an authoritative and streamlined academic tone throughout the text.

Round 2

Reviewer 1 Report

Comments and Suggestions for Authors

The author addressed most of the questions raised, though many experiments will need to be completed in the future. This is acceptable.